# An *O*-GlcNAcylomic Approach Reveals ACLY as a Potential Target in Sepsis in the Young Rat

**DOI:** 10.3390/ijms22179236

**Published:** 2021-08-26

**Authors:** Manon Denis, Thomas Dupas, Antoine Persello, Justine Dontaine, Laurent Bultot, Charlotte Betus, Thomas Pelé, Justine Dhot, Angélique Erraud, Anaïs Maillard, Jérôme Montnach, Aurélia A. Leroux, Edith Bigot-Corbel, Didier Vertommen, Matthieu Rivière, Jacques Lebreton, Arnaud Tessier, Michel De Waard, Luc Bertrand, Bertrand Rozec, Benjamin Lauzier

**Affiliations:** 1Université de Nantes, CHU Nantes, CNRS, INSERM, l’Institut du Thorax, F-44000 Nantes, France; manon.denis@univ-nantes.fr (M.D.); thomas.dupas@univ-nantes.fr (T.D.); antoine.persello@univ-nantes.fr (A.P.); charlotte.betus@etu.univ-nantes.fr (C.B.); thomas.pele@univ-nantes.fr (T.P.); justine.dhot@univ-nantes.fr (J.D.); angelique.erraud@univ-nantes.fr (A.E.); anais.maillard@inserm.fr (A.M.); jerome.montnach@univ-nantes.fr (J.M.); aurelia.leroux@oniris-nantes.fr (A.A.L.); michel.dewaard@univnantes.fr (M.D.W.); bertrand.rozec@univ-nantes.fr (B.R.); 2Pediatric Intensive Care Unit, CHU de Nantes, F-44000 Nantes, France; 3InFlectis BioScience, F-44000 Nantes, France; 4Université Catholique de Louvain, Institut de Recherche Expérimentale et Clinique, Pôle of Cardiovascular Research, B-1200 Brussels, Belgium; justine.dontaine@uclouvain.be (J.D.); laurent.bultot@uclouvain.be (L.B.); luc.bertrand@uclouvain.be (L.B.); 5Sanofi R&D, 1 Avenue Pierre Brossolette, F-44000 Chilly Mazarin, France; 6University Animal Hospital, Oniris Ecole Nationale Vétérinaire, Agroalimentaire et de l’Alimentation Nantes Atlantique, F-44000 Nantes, France; 7Departement of Biochemistry, CHU de Nantes, F-44000 Nantes, France; edith.bigot@univ-nantes.fr; 8Université Catholique de Louvain, de Duve Institute, Mass Spectrometry Platform, B-1200 Brussels, Belgium; didier.vertommen@uclouvain.be; 9Université de Nantes, CNRS, Chimie et Interdisciplinarité: Synthèse, Analyse, Modélisation (CEISAM), UMR CNRS 6230, Faculté des Sciences et des Techniques, F-44000 Nantes, France; matthieu.riviere@univ-nantes.fr (M.R.); jacques.lebreton@univ-nantes.fr (J.L.); arnaud.tessier@univ-nantes.fr (A.T.); 10WELBIO, B-1200 Brussels, Belgium

**Keywords:** *O*-GlcNAcylation, septic shock, children, mass spectrometry, *O*-GlcNAcylomics

## Abstract

Sepsis in the young population, which is particularly at risk, is rarely studied. *O*-GlcNAcylation is a post-translational modification involved in cell survival, stress response and metabolic regulation. *O*-GlcNAc stimulation is beneficial in adult septic rats. This modification is physiologically higher in the young rat, potentially limiting the therapeutic potential of *O*-GlcNAc stimulation in young septic rats. The aim is to evaluate whether *O*-GlcNAc stimulation can improve sepsis outcome in young rats. Endotoxemic challenge was induced in 28-day-old rats by lipopolysaccharide injection (*E. Coli* O111:B4, 20 mg·kg^−1^) and compared to control rats (NaCl 0.9%). One hour after lipopolysaccharide injection, rats were randomly assigned to no therapy, fluidotherapy (NaCl 0.9%, 10 mL·kg^−1^) ± NButGT (10 mg·kg^−1^) to increase *O*-GlcNAcylation levels. Physiological parameters and plasmatic markers were evaluated 2h later. Finally, untargeted mass spectrometry was performed to map cardiac *O*-GlcNAcylated proteins. Lipopolysaccharide injection induced shock with a decrease in mean arterial pressure and alteration of biological parameters (*p* < 0.05). NButGT, contrary to fluidotherapy, was associated with an improvement of arterial pressure (*p* < 0.05). ATP citrate lyase was identified among the *O*-GlcNAcylated proteins. In conclusion, *O*-GlcNAc stimulation improves outcomes in young septic rats. Interestingly, identified *O*-GlcNAcylated proteins are mainly involved in cellular metabolism.

## 1. Introduction

Sepsis is a dysregulated response of the host to an infectious pathogen whose response varies according to the pathogen and host. It is the result of a systemic inflammatory response syndrome (SIRS) following an infection [1,2,3,4]. Most studies focus on septic shock in adults, however, the populations most affected by septic shock are young children and elderly people [5,6,7,8]. Sepsis is an important but potentially preventable cause of pediatric death worldwide, with a mortality range from 4% to 50%. It causes 28% of mild disability in Europe [8]. Early identification, appropriate resuscitation and management are the key to optimizing outcomes for children with sepsis [9].

Unfortunately, a difficulty for the diagnosis of septic shock in pediatrics is related to the variability of physiological values according to age and pathophysiological characteristics. Unlike adults, in which septic shock is generally biphasic with an early phase of vasoplegia followed by a late phase of low cardiac output, a particular hemodynamic profile is observed in the younger population. In children, septic shock is a dynamic process with heterogeneous hemodynamic stages over the first 48 h [10]. Although the child’s pathophysiology is different (lower cardiac reserve, lower basal blood pressure), there is a general lack of pediatric studies. Many conclusions are transposed from the studies on adults. Although pediatric guidelines have recently been published [9], gaps in knowledge remain for these patients [11].

*O*-linked-N-acetyl glucosaminylation, more simply called *O*-GlcNAcylation (*O*-GlcNAc), is a post-translational modification consisting of a monosaccharide (β-d-N-acetylglucosamine) addition on serine or threonine residues [12]. This modification is regulated by a single pair of enzymes: the O-GlcNAc transferase (OGT) and the O-GlcNAcase (OGA) which add and remove the GlcNAc moiety, respectively [13]. Due to its implication in response to stress and cell survival, the *O*-GlcNAcylation is a potential new therapeutic strategy [14,15,16]. We and others have demonstrated that acute *O*-GlcNAc stimulation by pharmacological agents improves hemodynamic parameters and survival during sepsis in adult rats [17,18]. However, the *O*-GlcNAcylation levels vary throughout development with higher levels in the young rat [19]. In addition, recent studies showed that over stimulation of *O*-GlcNAcylation levels is associated with adverse effects [20]. According to these data, the promising results observed in adults cannot be directly transposed to children and further studies are needed before considering clinical trials. In this context, we evaluated the impact of *O*-GlcNAc stimulation in young rats and identified proteins involved in the process. We demonstrate that *O*-GlcNAc stimulation efficiently improves the pups’ response to endotoxemic shock and that ATP-citrate lyase (ACLY) represents a specific target of interest in this context.

## 2. Results

### 2.1. O-GlcNAcylation Levels Decrease during the Early Stage of Life

We have previously demonstrated that *O*-GlcNAcylation varies throughout aging [19]. Therefore, the first step was to evaluate *O*-GlcNAcylation levels right after weaning and compare them to adulthood. The global *O*-GlcNAcylation level significantly increased at D28 compared to D84 in the heart (D28/D84 ratio: 2.54 ± 0.16; *p* < 0.05) (Figure 1A). This reduction was associated with a significant decrease in the protein level of the two isoforms of glutamine fructose-6-phosphate amidotransferase (GFAT). Those enzymes are rate limiting enzymes of the hexosamine biosynthesis pathway which produces UDP-GlcNAc, the sugar donor for OGT. OGT decreased at D28 compared to D84 in the heart, while OGA was not significantly modified (*p* = 0.09) (Figure 1B). 

### 2.2. O-GlcNAcylation Levels Increase in Response to Sepsis in Young Rats

In adult rats (12 weeks of age), the *O*-GlcNAcylation levels remained stable during sepsis and the NButGT-mediated increase in *O*-GlcNAcylation levels during the early phase of septic shock was beneficial [17]. Similarly, in young rats, cardiac *O*-GlcNAcylation levels did not significantly vary between CTRL, LPS and LPS+R groups (Figure 1C). NButGT treatment induced a significant increase in *O*-GlcNAcylation levels, indicating that NButGT is also efficient at a younger age (*O*-GlcNAcylation levels relative to CTRL: LPS: 1.29 ± 0.07; LPS+R: 1.64 ± 0.08; NButGT: 2.46 ± 0.14; *p* < 0.05) (Figure 1C). In contrast to adult rats, the enzymes involved in *O*-GlcNAcylation are subject to significant variations in young rats. GFAT1 expression was increased in the LPS+R group compared to the LPS group, while GFAT2 expression was fivefold higher with LPS injection compared to CTRL. OGA decreased with LPS injection. NButGT treatment did not impact the expression of GFAT1, GFAT2 and OGA. OGT did not vary between the different groups (Figure 1D). 

### 2.3. O-GlcNAc Stimulation Significantly Improves Blood Pressure

Heart rate (HR) and blood pressure are key physiological parameters to ensure the animal’s general health during shock. In our study, the HR was similar in all groups (CTRL: 461 ± 9; LPS: 454 ± 7; LPS+R: 456 ± 7; NButGT: 478 ± 9; beats per minute; Figure 2A). Nevertheless, both basal systolic blood pressure (SBP) (Figure 2B) and mean arterial pressure (MAP) (Figure 2C) were lower after LPS injection. Our resuscitation protocol (LPS+R group) did not improve the SBP and MAP (+8 and +5 mmHg; *p* = 0.84 and *p* = 0.37, respectively) compared to the LPS group. Interestingly, NButGT restored values to those of the CTRL group (SBP: LPS+R: 74 ± 3; NButGT: 93 ± 4; mmHg; *p* < 0.05); (MAP: LPS+R: 55 ± 2; NButGT: 72 ± 4; mmHg; *p* < 0.05).

### 2.4. O-GlcNAc Stimulation in the Young Population Does Not Correct Circulating Parameters

The plasma lactate concentration was significantly increased in the LPS group compared to the CTRL group, in accordance with the septic shock definition. Three hours after shock induction in pups, neither fluid therapy nor NButGT treatment decreased lactates concentration (CTRL: 3.92 ± 0.25; LPS: 6.42 ± 0.45; LPS+R: 6.02 ± 0.34; NButGT: 6.34 ± 0.29; mmol·L^−1^; *p* < 0.05) (Figure 3A).

This hyperlactatemia is associated with metabolic acidosis (pH < 7.2). Endotoxemic shock results in a significant decrease in pH compared to the CTRL group. The pH is unchanged between the other studied groups (CTRL: 7.27 ± 0.02; LPS: 7.15 ± 0.02; LPS+R: 7.17 ± 0.13; NButGT: 7.18 ± 0.01; *p* < 0.05) (Figure 3B).

Hypoglycemia was observed in the LPS group compared to the CTRL group, and was restored neither by fluid therapy nor NButGT (CTRL: 11.67 ± 0.31; LPS: 2.68 ± 0.60; LPS+R: 3.05 ± 0.66; NButGT: 2.49 ± 0.47; *p* < 0.05; mmol·L^–1^; Figure 3C). Severe leukopenia was observed in the LPS group compared to the CTRL group and neither fluid therapy nor NButGT induced an improvement in leukocyte levels (CTRL: 5.67 ± 0.76; LPS: 1.83 ± 0.26; LPS+R: 1.79 ± 0.32; NButGT: 1.52 ± 0.32; *p* < 0.05; 10^3^ µL^–1^; Figure 3D).

Septic shock leads to multivisceral failure. Our LPS model resulted in a significant increase in troponin T, creatinine and ASAT compared to the control group. Fluid therapy treatment with or without NButGT did not improve markers of organ dysfunction (Figure 3E).

### 2.5. The General Condition and Survival of Young Animals Is Improved by Nbutgt Treatment

The Pediatric Risk of Mortality (PRISM) score used in pediatric intensive care units has been adapted for application to the animal. For this purpose, we have established threshold values based on the measured means and standard deviation of the control animals for the different parameters presented in Appendix A. This score considers the score for assessing the health status of the animals, based essentially on the observation of the animal’s behavior [17], hemodynamic variables (e.g., heart rate, arterial pressure) and circulating parameters (e.g., lactates, glycemia). As for patients, the higher the PRISM score, the worse the animal’s overall health.

To evaluate the impact of endotoxemic shock and treatments on health status, an adapted PRISM score was performed. Rats were monitored frequently throughout the study. The global health status was analyzed by a behavior score (A). Physiological functions and circulating parameters were measured (B). By summing these data, a PRISM score can be determined all animals. AHAS: Animal Health Assessment Score; SBP: systolic blood pressure; DBP: diastolic blood pressure; MAP: mean arterial pressure; HR: heart rate; RR: respiratory rate; ASAT: aspartate aminotransferase.

The PRISM score was increased in the LPS group compared to the CTRL group. Fluid therapy induced a limited improvement of the PRISM score compared to the LPS group. Rats in the NButGT group had a significantly lower PRISM score compared to the LPS+R group (CTRL: 2.1 ± 0.8; LPS: 15.8 ± 0.4; LPS+R: 13.2 ± 0.7; NButGT: 9.7 ± 0.6; *p* < 0.05; Figure 3F). 

Finally, NButGT resulted in a significant prolongation of survival time (NButGT: 36.00; LPS+R: 13.65; *p* < 0.001; median survival in hours; Figure 3G).

### 2.6. Identification of Putative Cardiac O-GlcNAcylated Proteins by O-GlcNAcylomic

Untargeted *O*-GlcNAcylomic mass spectrometry was performed on whole heart samples to identify putative cardiac *O*-GlcNAcylated proteins. This approach allowed us to identify 1327 putative cardiac *O*-GlcNAcylated proteins (Appendix A). With our quality criteria, 48 putative cardiac *O*-GlcNAcylated proteins were identified (Figure 4A,B and Appendix A). Among these proteins, only 33 are differentially *O*-GlcNAcylated amongst all the groups (Figure 4A,D). Through protein–protein interaction analysis, we highlighted that most of these proteins are involved in cardiac metabolism, structure and DNA-RNA processing (Figure 4C). Strikingly, only one protein, the ATP-citrate lyase (ACLY), is less *O*-GlcNAcylated after NButGT treatment (Figure 4B). Among the 33 proteins, we selected those not described in the literature and with a major role in heart structure or function, for further study. After applying this last filter, two proteins were selected to be more closely studied: troponin C (Tnnc1) and mitochondrial 2-oxoglutarate/malate carrier protein (Slc25a11). The Log_2_FC profil of the three protein of interest is showed with black arrow.

While protein expression of troponin C remained constant in all the condition studied (Figure 5A-ii), the level of *O*-GlcNAcylation was increased in the LPS group, reduced with fluidotherapy and increased with NButGT treatment (Figure 5A-i). Western blot analysis of Slc25a11 revealed a lower expression in the LPS group and no variation in the LPS+R and NButGT group (Figure 5B-ii). Interestingly, *O*-GlcNAcylation of Slc25a11 follows a completely different pattern and was increased in the LPS group, reduced with fluidotherapy and increased with NButGT treatment (Figure 5B-i). Taken together, these results highlight the potential role of O-GlcNAcylation of these proteins during shock. The ATP-citrate lyase is less O-GlcNAcylated in the NButGT group (Figure 5C-i). To confirm that the observed variation of *O*-GlcNAcylation was not due to changes in protein expression, we analyzed ACLY total protein levels. Interestingly, the protein expression remained stable in all groups (Figure 5C-ii).

### 2.7. ATP-Citrate Lyase Is Less O-GlcNAcylated in the NButGT Group

In our set of proteins, the ATP-citrate lyase is the only protein with a reduction in *O*-GlcNAcylation level in the NButGT group (Figure 4B).

Serine and threonine are the two amino acids potentially *O*-GlcNAcylated and/or phosphorylated. The phosphorylation of ACLY on threonine 447 and serine 451 is unchanged with LPS challenge (Figure 6A) while the phosphorylation of serine 455 is increased with LPS injection (Figure 6B). Strikingly, neither fluidotherapy nor NButGT treatment affected these phosphosites (Figure 6A,B).

## 3. Discussion

Our study focused on the impact of septic shock treatment in young animals. We found that, despite higher O-GlcNAcylation levels in young rats, increasing these levels is an interesting strategy to improve blood pressure and time of survival in septic shock. In addition, using a mass spectrometry approach, we identified proteins involved in metabolism that are differentially *O*-GlcNAcylated in our treated condition.

### 3.1. Cardiac O-GlcNAc throughout Aging

We first confirmed that *O*-GlcNAcylation levels are higher in cardiac tissues from young compared to adult rats. Our previous results demonstrated that *O*-GlcNAcylation levels vary in a tissue- and time-specific manner, suggesting that *O*-GlcNAcylation levels may play an important role in development [19]. This result challenges the current paradigm that associates prolonged high levels of *O*-GlcNAcylation with deleterious effects [21]. Indeed, a previous study demonstrated that overstimulation of *O*-GlcNAcylation levels may be associated with adverse effects in a brain infarct model [20].

### 3.2. Validation of the Septic Shock Model in Young Rats

Using several shock models, we have previously shown that increasing *O*-GlcNAcylation levels improve the health status of adult rats [17]. In order to validate these observations in young rats, it was necessary to adapt the model to 28-day-old rats. As expected, based on the definition of septic shock, the injection of LPS causes hypotension and filling-resistant increase in lactate associated with metabolic acidosis. However, unlike adult rats, young rats do not exhibit tachycardia. Young rats also have limited cardiac reserve and may not benefit hemodynamically from an increased heart rate in the same manner as adults do. Resting HR of young rats is higher than in adults and tachycardia may not allow for adequate diastolic filling [11,22]. As previously described, in the young population the heart decompensates quickly with sustained bradycardia and drop of cardiac output [22]. This observation is in accordance with results provided in a neonatal septic shock model in pigs, showing an early increase in HR, followed by a rapid decrease in HR associated with a decrease in MAP prior to death [23]. Similarly, a high mortality rate is observed early in our model. This model of septic shock in young rats could be considered as representative of the clinical symptomatology of pediatric septic shock.

### 3.3. Impact of O-GlcNAcylation Stimulation in Septic Shock in Young Rats

To evaluate the impact of our treatment on the proteins involved in the regulation of O-GlcNAc levels, GFAT1, GFAT2 (the rate limiting enzyme of the hexosamine biosynthetic pathway), OGA and OGT were evaluated by Western blot. As described in the literature, GFAT2 expression is increased in response to LPS in all groups and NButGT treatment does not significantly reduce its expression [24]. Interestingly, and as previously described, NButGT treatment increases O-GlcNAc levels without affecting enzyme expression [17]. However, when compared to the LPS+R group, GFAT1 expression tends to decrease (*p* = 0.07) in the NButGT group, potentially indicating a negative feedback on its expression.

In this study, we have demonstrated that NButGT treatment leading to increasing O-GlcNAcylation levels (~1.5 fold) and restoring SBP and MAP is effective in young rats despite physiologically higher O-GlcNAcylation levels. NButGT treatment also tends to increase heart rate, which could compensate for the low cardiac reserve. However, NButGT does not reduce lactatemia and correct metabolic acidosis. The new definition of septic shock includes the notion of multiple organ failure, which is found in this model with heart, kidney and liver failure [9]. NButGT treatment does not decrease cardiac troponin T (Tc) and creatinine levels as we previously reported in adult rats [17]. However, the increase in blood troponin Tc and creatinine levels is smaller in young rats. In a mice model of LPC, despite equivalent cardiac troponin I and T levels, adult mice exhibited more severe cardiac damage than young mice [25]. The relevance of an increase in troponins is currently discussed in children in sepsis [26,27]. Finally, our study demonstrated that NButGT improves the adapted PRISM score used by clinicians as a predictive marker of mortality and decreases mortality as previously described in adult rats [17]. These results are supported by a recent study in pediatric intensive care units in which a low glutamine level in plasma (essential for the formation of UDP-GlcNAc via the hexosamines biosynthesis pathway) (<420 μmol/L) at admission is associated with an increased risk of multivisceral failure [28]. In addition, glutamine supplementation tends to reduce morbidity and mortality following sepsis [29].

Our results show for the first time that a difference in O-GlcNAcylation levels between the basal levels and the post-stimulation levels is more beneficial in stress situations than intrinsic O-GlcNAcylation levels by themself. Thus, an early treatment to increase the levels of O-GlcNAcylation of proteins at any basal level may be effective.

### 3.4. O-GlcNAcylomic Analysis

To date, no studies have evaluated the *O*-GlcNAcylome during septic shock in young rats or humans. *O*-GlcNAcylomic analysis allowed us to identify new cardiac *O*-GlcNAcylated proteins. We focused our attention on three proteins: Tnnc1, Slc25a11 and ACLY.

Troponin complex (troponin T, C and I) is a key modulator of muscle contraction through interaction with tropomyosin and inhibition of the ATPase activity of the actomyosin complex [30]. Sepsis has been associated with a decrease in contractility [31] and an increase in *O*-GlcNAcylation of troponin Tc has been reported in a rat model of myocardial infarction [32]. Stimulation of *O*-GlcNAcylation of cardiac troponin I has been associated with a decrease in contractility [32]. For the first time, via untargeted mass spectrometry, we have identified troponin C as being *O*-GlcNAcylated in the heart. The functional impact of O-GlcNAcylation on troponin C remains to be explored.

In addition to cardiac contraction, septic shock also affects cardiac metabolism. The mitochondrion, a key organelle in metabolism, is affected during sepsis and is involved in the multiple organ dysfunction [33]. Slc25a11 is a mitochondrial solute carrier involved in several pathways among which are gluconeogenesis from lactate, the oxoglutarate-isocitrate shuttle and the malate-aspartate shuttle [34,35]. Slc25a11 has also been described to be involved in apoptosis and insulin secretion [36,37]. Acute stimulation of *O*-GlcNAcylation of mitochondrial proteins has been associated with cardioprotective effects [38,39]. Deciphering the impact of *O*-GlcNAcylation on Slc25a11 may lead to a better understanding of the impairment of mitochondrial metabolism during sepsis.

ATP-citrate lyase (ACLY) is responsible for the conversion of citrate to acetyl-CoA and oxaloacetate [40]. We reported here a unique *O*-GlcNAcylation profile of ACLY. The *O*-GlcNAcylation level decreases when OGA is inhibited by NButGT. This particular *O*-GlcNAc signature may be the proof of the key role played by *O*-GlcNAcylation of ATP-citrate lyase in response to metabolic change and stress. Recently, ATP-citrate lyase inhibition has been proposed as a potential treatment track for cancer and/or metabolic diseases [41].

Interestingly, we found that phosphorylation of serine 455 of ACLY, which is an activating phosphorylation [42], is increased in the LPS group. This result is supported by recent studies that demonstrated that LPS-mediated activation of ATP-citrate lyase is essential for the macrophage inflammatory response via nitric oxide, reactive oxygen species, prostaglandin E2 inflammatory mediators′ production [43] and inflammatory gene induction [42]. More recently, an increase in ACLY has been reported specifically in children surviving sepsis [44]. All these data pinpoint the involvement of ATP citrate lyase in sepsis and open a new therapeutic avenue. 

## 4. Limits

Our study was carried out on animals aged 28 days, which corresponds to the end of weaning in rats. However, we have studied only this specific period of time while we have shown that *O*-GlcNAcylation levels are highly variable with age [19]. We cannot exclude that the observed effects would be different at younger (neonatal) or older ages. In our LPS model, treatments (fluidotherapy and NButGT) were administered subcutaneously because the penile vein is not accessible at 28 days unlike the 84-day study. The intravenous route allows a faster administration and diffusion of the active ingredients, which could improve the beneficial effects. 

## 5. Conclusions

Our data suggest that increasing *O*-GlcNAcylation in the early phase of septic shock is an interesting therapeutic strategy to reduce the septic shock burden for children. Further studies are necessary to confirm its putative therapeutic interest and the follow-up of *O*-GlcNAcylation levels in patients will be necessary to consider designing new treatment in clinic. Proteins identified in mass-spectrometry and more specifically ACLY have to be explored extensively to understand the role of acute *O*-GlcNAc stimulation in stress response.

## 6. Methods

Additional details on the animal model, tissue preparation and methods used are provided in an online data supplement.

### 6.1. Reagents

O-GlcNAcase inhibitor NButGT was synthesized using Matthew S. Macauley methods [45].

### 6.2. Animal Model and Measures

Rats were housed under standard conditions of temperature (21–24 °C), humidity (40–60%) and 12 h light/dark cycle with light period starting at 07:00 a.m. Food and water were available ad libitum.

Thirty minutes before shock induction, rats received intravenous buprenorphine (0.03 mg/kg). Rats were anesthetized with an O_2_–/isoflurane mixture (induction: 5% isoflurane, flow rate 1L/min; maintenance: 2% isoflurane, flow rate 0.5 L/min). Endotoxemic shock was induced in twenty-eight-day-old male Wistar rats (Charles River, Saint-Germain-Nuelles, France) by lipopolysaccharides (LPS) injection (*E. Coli* O111:B4, 20 mg·kg^−1^) (Sigma-Aldrich, Saint Louis, MI, USA) and was compared to control rats (injection of NaCl 0.9%—CTRL). One hour after LPS injection, rats were randomly assigned to: no therapy (LPS), fluidotherapy (NaCl 0.9%, 10 mL·kg^–1^ – LPS+R) ± NButGT (NButGT, 10 mg·kg^−1^—NButGT) to increase *O*-GlcNAcylation levels (*n* = 11 per group). Two hours later, physiological functions, blood gas parameters and plasmatic markers of sepsis severity were measured, enabling us to calculate an adapted Pediatric Risk of Mortality score (PRISM score) (Appendix A). Animals were euthanized with a lethal dose of pentobarbital (Dolethal^®^, Vetoquinol, Paris, France) and hearts were immediately freeze-clamped to preserve post-translational modifications. In another group, the impact of treatment was evaluated on survival (*n* = 16 per group). During this period, animals were prematurely euthanized with a lethal dose of pentobarbital if they met specific criteria: incapacity to move, decubitus position, difficulty in breathing.

### 6.3. Tissue Preparation and Protein Extraction

Frozen hearts were crushed to obtain a powder as previously described [19] (Appendix A).

### 6.4. Western Blot

Western blotting experiments were performed as previously described [46] (Appendix A). Analysis was performed using Image Lab software (Image Lab 6.1, Bio-Rad, CA, USA). ^#^: dilutions carried out in 3% BSA; *: dilutions carried out in 5% milk.

### 6.5. Proteomic Study

Proteomic analyses were performed as previously described [19]. Peptides were identified and quantified by HR/AM LC-MS/MS on an Orbitrap Tribrid Fusion Lumos as described [47]. The relative abundances of 5593 putative *O*-GlcNAcylated proteins were evaluated by label-free quantification within Proteome Discoverer from MS1 intensities (Appendix A). Profiles of abundances were analyzed using RStudio software (RStudio 3.12.0, RStudio Team (2020). RStudio: Integrated Development for R. RStudio, PBC, Boston, MA, USA) and protein–protein networks were analyzed using STRING database (v11) [48].

### 6.6. Statistical Analyses

Results were expressed as an average ± SEM of n different rats. Analyses of Western blots were expressed in relation to the average of the stain free and then reduced to the average of the control samples (D84 or CTRL). For D28-D84 Western blots, data were analyzed by a Mann–Whitney test. For CTRL-LPS-LPS+R-NButGT Western blots, physiological and biological parameters and PRISM score, data were analyzed by a Kruskal–Wallis test. Survival analysis is presented using a Kaplan–Meyer curve and was evaluated using a Mantel–Cox test. A value of *p* < 0.05 was considered significant. All statistical calculations and graphs (except those performed with R software) were performed using GraphPad PRISM software (version 7.00).

## Figures and Tables

**Figure 1 ijms-22-09236-f001:**
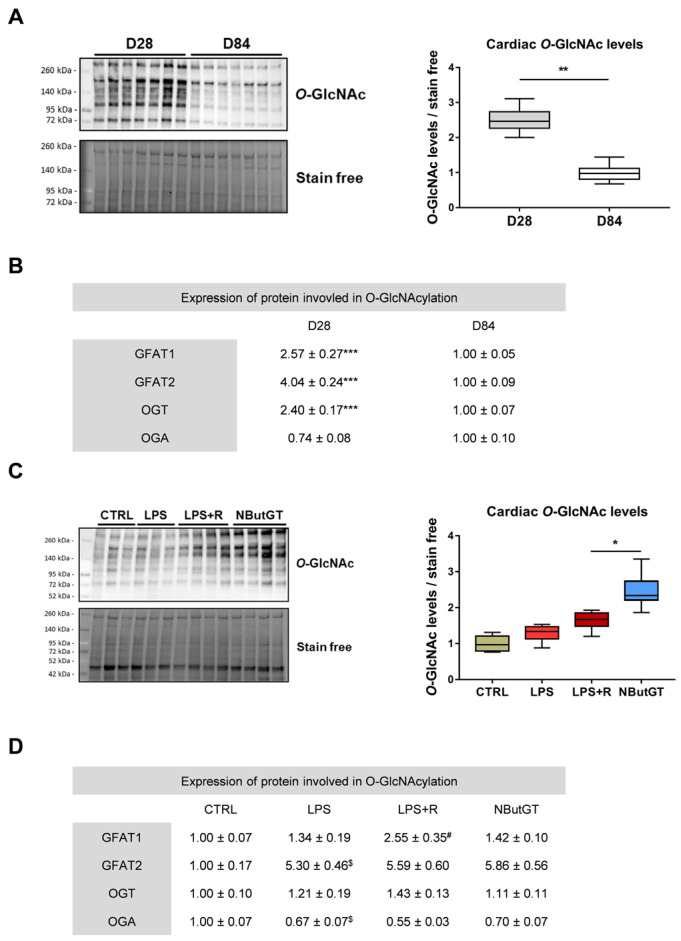
Impact of aging and septic shock on *O*-GlcNAcylation. Evaluation by Western blot of O-GlcNAcylation levels of cardiac proteins (**A**) and involved protein levels (**B**) in animals aged 28 and 84 days (D28 (young) and D84 (adult). Statistical significance was assessed by Mann–Whitney test (**: *p* < 0.01; ***: *p* < 0.001). Evaluation by Western blot of O-GlcNAcylation levels of cardiac proteins (**C**) and involved protein levels (**D**) in CTRL, LPS, LPS+R and NButGT group. Statistical significance was assessed by Kruskal–Wallis test with uncorrected Dunn’s post-test (*: *p* < 0.05; ^#^: *p* < 0.05 vs. LPS; ^$^: *p* < 0.05 vs. CTRL). Quantification was performed in relation to stain free. Results are expressed as mean ± SEM. CTRL: control group, LPS: i.v. injection of LPS (20 mg/kg), LPS+R: subcutaneous administration of 10 mL/kg of NaCl 0.9%, NButGT: resuscitation supplemented with NButGT (10 mg/kg). *n* = 7–9.

**Figure 2 ijms-22-09236-f002:**
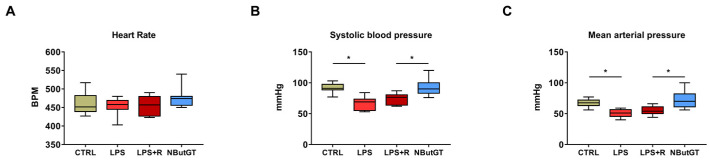
Impact of endotoxemic shock and O-GlcNAc stimulation on heart rate, systolic blood pressure and mean arterial pressure in young rats. To study the impact of endotoxemic shock and treatments, measures of heart rate (**A**), systolic blood pressure (**B**) and mean arterial pressure (**C**) were performed. Statistical significance was assessed by Kruskal–Wallis test with uncorrected Dunn’s post-test (*: *p* < 0.05). CTRL: control group, LPS: i.v. injection of LPS (20 mg/kg), LPS+R: subcutaneous administration of 10 mL/kg of NaCl 0.9%, NButGT: resuscitation supplemented with NButGT (10 mg/kg). Results are expressed as mean ± SEM. *n* = 9–11.

**Figure 3 ijms-22-09236-f003:**
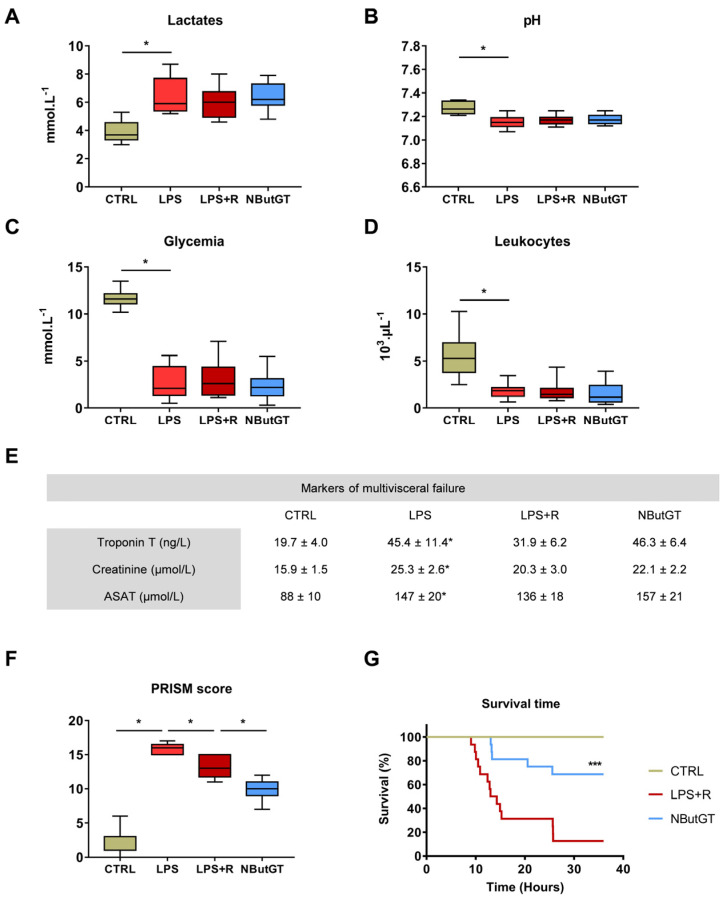
Impact of endotoxemic shock and *O*-GlcNAc stimulation on circulating parameters, adapted PRISM score and survival time. To study the impact of endotoxemic shock and treatments, the levels of lactates (**A**), pH (**B**), glycemia (**C**), leukocytes (**D**) and markers of organ function (Troponin T, Creatinine and Aspartate transaminase (ASAT) (**E**) were measured in venous blood collected from rats anesthetized with isoflurane. Statistical significance was assessed by Kruskal–Wallis test with uncorrected Dunn’s post-test (*: *p* < 0.05, relative to CTRL). (**F**) Adapted PRISM score was realized with behavior score, physiological and circulating parameters. Statistical significance was assessed by Kruskal–Wallis test with uncorrected Dunn’s post-test (*: *p* < 0.05). *n* = 5–7. (**G**) Survival analysis is presented using a Kaplan–Meyer curve. Statistical significance was assessed by Mantel–Cox test (***: *p* < 0.001, relative to LPS+R). *n* = 16 per group. CTRL: control group, LPS: i.v. injection of LPS (20 mg/kg), LPS+R: subcutaneous administration of 10 mL/kg of NaCl 0.9%, NButGT: resuscitation supplemented with NButGT (10 mg/kg).

**Figure 4 ijms-22-09236-f004:**
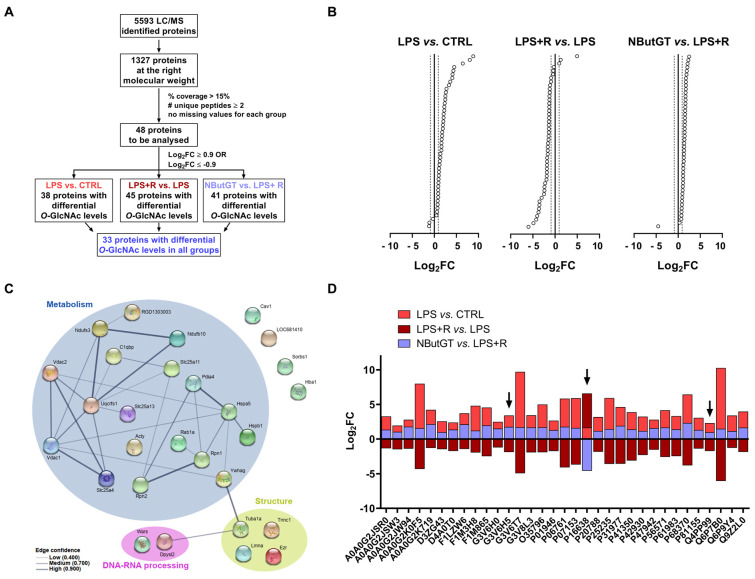
Flow chart and proteins of interest. (**A**) Flowchart of mass spectrometry analysis (**B**) Log_2_(fold change) of *O*-GlcNAcylation levels of selected proteins. (**C**) Protein-protein network analysis (STRING network) of proteins with a Log_2_(fold change) >0.9 or <−0.9. (**D**) Log_2_(fold change) of *O*-GlcNAcylation levels of proteins differentially *O*-GlcNAcylated in the LPS, LPS+R and NButGT groups. CTRL: control group, LPS: i.v. injection of LPS (20 mg/kg), LPS+R: subcutaneous administration of 10 mL/kg of NaCl 0.9%, NButGT: resuscitation supplemented with NButGT (10 mg/kg). *n* = 2.

**Figure 5 ijms-22-09236-f005:**
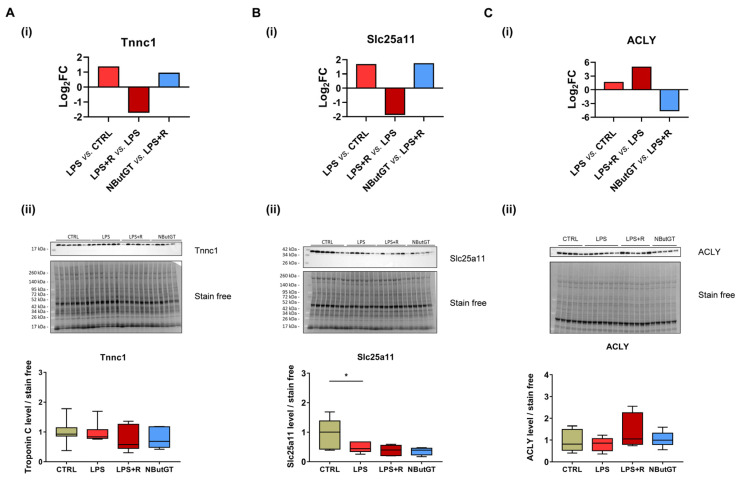
*O*-GlcNAcylation and expression of troponin C, mitochondrial 2-oxaloacetate/malate transporter protein and ATP-citrate lyase. Log_2_(fold change) of *O*-GlcNAcylation of troponin C (**A-i**), mitochondrial 2-oxaloacetate/malate transporter protein (**B-i**) and ATP-citrate lyase (**C-i**) between different groups. *n* = 2. Evaluation by Western blot of the expression of troponin C (**A-ii**), mitochondrial 2-oxaloacetate/malate transporter protein (**B-ii**) and ATP-citrate lyase (**C-ii**) in different groups. Statistical significance was assessed by Kruskal–Wallis test with uncorrected Dunn’s post-test (*: *p* < 0.05). Quantification was in relation to stain free. Results are expressed as mean ± SEM. CTRL: control group, LPS: i.v. injection of LPS (20 mg/kg), LPS+R: subcutaneous administration of 10 mL/kg of NaCl 0.9%, NButGT: resuscitation supplemented with NButGT (10 mg/kg). *n* = 6–7.

**Figure 6 ijms-22-09236-f006:**
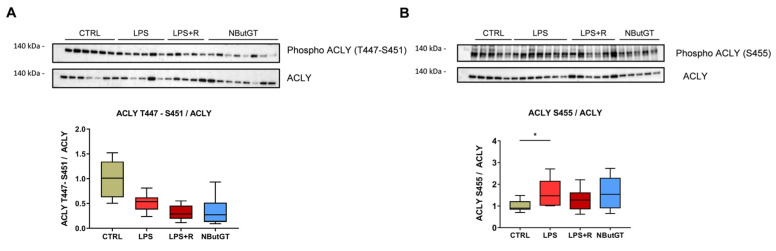
Impact of septic shock and NButGT treatment on ACLY phosphorylation. Evaluation by Western blot of the phosphorylation of ACLY on threonine 447, serine 451 (**A**) and serine 455 (**B**). Statistical significance was assessed by Kruskal–Wallis test with uncorrected Dunn’s post-test (*: *p* < 0.05). Results are expressed as mean ± SEM. CTRL: control group, LPS: i.v. injection of LPS (20 mg/kg), LPS+R: subcutaneous administration of 10 mL/kg of NaCl 0.9%, NButGT: resuscitation supplemented with NButGT (10 mg/kg). *n* = 5–9.

## Data Availability

The data that support the findings of this study are available from the corresponding author upon reasonable request.

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
