# Peer review of "An O-GlcNAcylomic Approach Reveals ACLY as a Potential Target in Sepsis in the Young Rat"

_ijms, 2021, doi:10.3390/ijms22179236_

Round 1

Reviewer 1 Report

I have found the proposed manuscript interesting and can attract readers of the International Journal of Medical Sciences. As described in the Introduction section, septic shock is a life-threatening condition, especially in a young population. Furthermore, its treatment strategy differs from that administered in adults. According to my understanding and knowledge, an in vitro model using rats and applied in the study seems to be adequate to perform the research on potential OGlcNAcylation stimulating agents as a novel strategy for septic shock treatment. However, in my opinion, there can be addressed few concerns and necessary improvements, which possibly would increase the scientific value of presented studies.

According to the Instruction for Authors, the manuscript's main body does not follow the general structure dedicated to the Research paper.

In my opinion, there is a missing Kaplan-Mayer survival curve under LPS treatment in Fig. 3.

All figures need to be improved regarding quality and resolution. The details are hardly visible in the present form.

The discussion section needs to be restructured, including the following sections from 4 to 6 into one Discussion paragraph with appropriate sub-headings.

The supplementary materials should follow the sections and subsections of the manuscript's main body. All figures and tables in supplementary material should also be placed immediately after the first citation to make it easier to follow. Moreover, I could not find Table E3. Is it referring to another paper by Merlet et al. 2013 or this manuscript?

Author Response

Point 1: According to the Instruction for Authors, the manuscript's main body does not follow the general structure dedicated to the Research paper.

 Response 1: We thank the reviewer for his comment. The manuscript’s main body have been changed following IJMS instructions.

Point 2: In my opinion, there is a missing Kaplan-Mayer survival curve under LPS treatment in Fig. 3.

Response 2: Author would like to thanks the reviewer for this comment, unfortunately we did not get the legal authorisation from our ethic comity to run a mortality curve on our LPS group as “the resuscitation procedure is mandatory in clinical practice and no sepsis patient will ever stay untreated”. We are aware that this represents a potential limitation, but we cannot run this experiment without the authorisation of the ethic comity. We had a similar comment in a previous paper published in 2019 (doi: 10.1038/s41598-019-55381-7) and we were not able to respond as well. We are sorry for this limitation.

Point 3: All figures need to be improved regarding quality and resolution. The details are hardly visible in the present form.

Response 3: Authors are sorry for this inconvenience; figure quality has been improved.

Point 4: The discussion section needs to be restructured, including the following sections from 4 to 6 into one Discussion paragraph with appropriate sub-headings.

Response 4: The section has been reworked and should now show correctly.

Point 5: The supplementary materials should follow the sections and subsections of the manuscript's main body. All figures and tables in supplementary material should also be placed immediately after the first citation to make it easier to follow. Moreover, I could not find Table E3. Is it referring to another paper by Merlet et al. 2013 or this manuscript?

Response 5: According to reviewer suggestion, supplementary materials have been reworked and should now be easier to follow. The authors made an error in the title of the document and apologize for this inconvenience. The Table E3 is a supplementary material present in this manuscript.

Reviewer 2 Report

This study conducted both proteomic and physiological assays and suggested that promoting O-GlcNacylation level in the early phase of septic shock might serve as a novel therapeutic strategy. ACLY was proposed as a promising candidate in targeting therapy of sepsis in young population.

Comments:

Pros:

  1. Shock model was successfully adapted on young rats from adult rats with a series of validations.
  2. Physiological assays including test of blood pressure and all four circulating parameters exhibited significant and consistent alteration as expected or previously reported.
  3. Combination of proteomic profiling and physiological test gives rise to correlation between phenotype and molecular mechanism.

Cons:

  1. Lack of explanation regarding the differential expression of GFAT1, GFAT2, OGT and OGA in young rats with/without sepsis (Figure 1D).
  2. It still unknown why higher O-GlcNacylation improved PRISM score without affecting any circulating parameters or biomarkers (Figure 2F).
  3. Lack of an important control (LPS) in survival time (Figure 2G).
  4. In differential expression analysis of proteomic data, three pairs of simple fold change comparison were performed, which amplify the FDR by three times and does not show any statistical significance.
  5. In standard PTM analysis with mass spectrometry, the flow through after enrichment is subjected to proteome analysis and used as correction at protein-level. A gel based assay on distinct samples in this study involved a lot of false discoveries.
  6. All three candidate proteins were cytosol proteins but O-GlcNacylation in most cases happens in nuclear fraction.
  7. Lack of solid and reasonable evidence in regarding ACLY as a potential therapeutic target.

Author Response

Response to Reviewer 2 Comments

Point 1: Lack of explanation regarding the differential expression of GFAT1, GFAT2, OGT and OGA in young rats with/without sepsis (Figure 1D).

 Response 1: Author would like to thank the reviewer for pointing out this omission. A sentence has been added in the manuscript to explain why we evaluated GFAT 1 - 2, OGT and OGA proteins. The corresponding discussion section now contains the following sentences:

“To evaluate the impact of our treatment on the proteins involved in the regulation of O-GlcNAc levels, GFAT1, GFAT2 (the rate limiting enzyme of the hexosamine biosynthetic pathway), OGA and OGT were evaluated by western blot. As described in the literature, GFAT2 expression is increased in response to LPS in all groups and NButGT treatment does not significantly reduce it expression [24]. Interestingly, and as previously described, NButGT treatment increases O-GlcNAc levels without affecting enzyme expression [17]. However, when compared to the LPS+R group, GFAT1 expression tends to decrease (p = 0.07) in the NButGT group, potentially indicating a negative feedback on its expression.”

Point 2: It still unknown why higher O-GlcNacylation improved PRISM score without affecting any circulating parameters or biomarkers (Figure 2F).

Response 2: We thank the reviewer for this comment. PRISM score is adapted from clinic and is a multifactorial score including behaviour (drinking, feeding…), physiological parameters (heart rate, respiration rate, arterial blood pressure…), and circulating parameters (glycemia, lactate…). Hence even if the circulating parameters are identical, if the pups drink, eat and move correctly in the cage the score will be lower. Moreover, the blood pressure was significantly improved in the NButGT group which will also reduce the PRISM score.

As the physiological parameters were not mentioned in the manuscript and it was misleading, the legend of the figure 3 now show:

“Adapted PRISM score was realized with behavior score, physiological and circulating parameters”

Point 3: Lack of an important control (LPS) in survival time (Figure 2G).

Response 3: We thank the reviewer for this comment. As explained to reviewer 1, we did not get the legal authorisation from our ethic comity to run a mortality curve on our LPS group as “the resuscitation procedure is mandatory in clinical practice and no sepsis patient will ever stay untreated”. We are aware that this represents a potential limitation, but we cannot run this experiment without the authorisation of the local ethic comity. We had a similar comment in a previous paper published in 2019 (doi: 10.1038/s41598-019-55381-7) and we were not able to respond as well. We are sorry for this limitation.

Point 4: In differential expression analysis of proteomic data, three pairs of simple fold change comparison were performed, which amplify the FDR by three times and does not show any statistical significance.

Response 4: We thank the reviewer for this comment. In a first step, the FDR is calculated for each peptide/protein identification with a cutoff at 0.05. A selection is thus already made for each of the 3 analyses (LPS vs. CTRL, LPS+R vs. LPS and NButGT vs. LPS+R). In a second step, only proteins with our criteria of quality (coverage > 15%, # unique peptides ≥ 2 and no missing values) and with a value of Log2FC ≥ 0.9 or ≤ - 0.9 have been selected and compared. As mentioned by the reviewer, no statistical analysis was performed on the proteomic study. The goal was to select candidates of interest and study them specifically afterwards.

Point 5: In standard PTM analysis with mass spectrometry, the flow through after enrichment is subjected to proteome analysis and used as correction at protein-level. A gel based assay on distinct samples in this study involved a lot of false discoveries.

Response 5: We thank the reviewer for this comment. In our study, O-GlcNAcylated proteins are immunoprecipitated in RIPA buffer (25 mM Tris-HCL, 150 mM NaCl, 0.1% SDS, 1% NP40, 1% Sodium Deoxycholate, 1µM PUGNAc, 1 µM Alloxan). The flow through after enrichment is therefore in a buffer incompatible with MS analysis. Moreover, this buffer also contains a large excess of antibodies used for the IP which will interfere with protein quantification by MS. As you mentioned by the reviewer, because of the risk of false discoveries, protein of interest (such as ACLY, Troponin C and Slc25a11) where more extensively studied and their level of expression analyzed by western-blotting.

Point 6: All three candidate proteins were cytosol proteins but O-GlcNacylation in most cases happens in nuclear fraction.

Point 7: Lack of solid and reasonable evidence in regarding ACLY as a potential therapeutic target.

Response 7: Author agree with reviewer comment and that is why the conclusion section state:

“Proteins identified in mass-spectrometry and more specifically ACLY have to be explored extensively to understand the role of acute O-GlcNAc stimulation in stress response.”

The title is made to point out the fact that ACLY is the only protein with a lower O-GlcNAc level in our study beside NButGT treatment. We would be willing to change the title if requested by the reviewer.

Round 2

Reviewer 2 Report

The authors have addressed my questions/comments.